# Distribution and abundance of azaspiracid-producing dinophyte species and their toxins in North Atlantic and North Sea waters in summer 2018

Stephan Wietkamp[1], Bernd Krock[1], Dave Clarke[2], Daniela Voß[3], Rafael Salas[2], Jane Kilcoyne[2], Urban Tillmann[1]*

1 Alfred-Wegener-Institute, Helmholtz Centre for Polar and Marine Research, Bremerhaven, Germany, 2 Marine Institute, Galway, Ireland, 3 Institute for Chemistry and Biology of the Marine Environment (ICBM), Carl von Ossietzky Universität Oldenburg, Wilhelmshaven, Germany

* urban.tillmann@awi.de

**Data Availability Statement:** Cruise CTD data are available at https://doi.org/10.1594/PANGAEA.

## Abstract

Representatives of the marine dinophyte family Amphidomataceae produce lipophilic phycotoxins called azaspiracids (AZA) which may cause azaspiracid shellfish poisoning (AZP) in humans after consumption of contaminated seafood. Three of the four known toxigenic species are observed frequently in the eastern North Atlantic. In 2018, a research survey was performed to strengthen knowledge on the distribution and abundance of toxigenic Amphidomataceae and their respective toxins in Irish coastal waters and in the North Sea. Species-specific quantification of the three toxigenic species (*Azadinium spinosum*, *Azadinium poporum* and *Amphidoma languida*) was based on recently developed qPCR assays, whose performance was successfully validated and tested with specificity tests and spike experiments. The multi-method approach of on-board live microscopy, qPCR assays and chemical AZA-analysis revealed the presence of Amphidomataceae in the North Atlantic including the three targeted toxigenic species and their respective AZA analogues (AZA-1, -2, -33, -38, -39). *Azadinium spinosum* was detected at the majority of Irish stations with a peak density of $8.3 \times 10^4$ cells $L^{-1}$ and AZA (AZA-1, -2, -33) abundances up to 1,274 pg $L^{-1}$. *Amphidoma languida* was also present at most Irish stations but appeared in highest abundance in a bloom at a central North Sea station with a density of $1.2 \times 10^5$ cells $L^{-1}$ and an AZA (AZA-38, -39) abundances of 618 pg $L^{-1}$. *Azadinium poporum* was detected sporadically at the Irish south coast and North Sea and was rather low in abundance during this study. The results confirmed the wide distribution and frequent occurrence of the target species in the North Atlantic area and revealed, for the first time, bloom abundances of toxigenic Amphidomataceae in this area. This emphasizes the importance of future studies and monitoring of amphidomatacean species and their respective AZA analogues in the North Atlantic.

896405. All other relevant data are within the paper and its Supporting Information files

**Funding:** This work was supported by funding of the German Ministry for Education and Research (project RIPAZA, 03F0763A) and by the PACES II research program of the Alfred-Wegener-Institute as part of the Helmholtz Foundation initiative in Earth and Environment.

**Competing interests:** The authors have declared that no competing interests exist.

## Introduction

Marine toxic microalgae are of concern for human health and aquaculture industries worldwide. Higher cell densities of harmful algae may occur at any time of the year, when conditions are favorable, but the complex interplay of both biotic and abiotic factors determining these outbreaks are still not fully understood [1, 2]. These harmful algal blooms can cause the death of aquatic organisms, including large fish-kills, or may lead to accumulation of toxic compounds within the aquatic food web. Elevated phycotoxin levels have the potential to cause human intoxications after consumption of contaminated seafood and therefore can have negative economic and health impacts on a global scale [3].

Since shellfish (e.g., oysters, blue mussels) are becoming increasingly important as a nutritious and sustainable food source for humans, a large number of studies have focused on toxins produced by certain phytoplankton species, which accumulate in shellfish. Well-known examples of those toxic algal compounds are yessotoxins (YTX, produced by e.g., *Protoceratium reticulatum*), diarrhetic-shellfish-poisoning toxins (DSP, produced by e.g., *Dinophysis* spp.) and paralytic-shellfish-poisoning toxins (PSP, produced by e.g., *Alexandrium* spp.) [4–7]. While these toxins have been thoroughly studied since the 1970s and 80s, another group of microalgal toxins, azaspiracids (AZA), which were more recently discovered [8], are now known to also accumulate in shellfish. These lipophilic polyether toxins are responsible for the so-called azaspiracid-shellfish-poisoning (AZP) syndrome, which is characterized by several gastrointestinal symptoms in humans after consumption of contaminated shellfish [9]. In 2002, the EU set a regulatory limit of 160 µg kg$^{-1}$ AZA equivalents (AZA-1, -2 and -3) in whole shellfish, or any edible part. AZA contamination is a major problem for shellfish producers in Ireland, where AZA levels above the threshold have been recurrently reported [10, 11] in a number of species of bivalve molluscan shellfish including mussels, oysters, clams and cockles. The regulatory limit has also been exceeded in Norway in 2002/2003, leading to first closures of mussel farms at the entire south coast due to AZA contamination [12]. More recently, AZA were found, for the first time, along the Atlantic coast of southern Spain with elevated levels observed in molluscan shellfish [13].

It took 10 years after the structural elucidation of AZA-1 [8] until a source organism of AZA was identified, using on-board microscopy and LC-MS/MS (Liquid-Chromatography coupled with tandem Mass Spectrometry) [14]. The causative organism was later described as *Azadinium spinosum*, a new dinophyte species within a newly erected genus [15]. Since then, two other AZA producing species of *Azadinium* have been described i.e., *Az. poporum* and *Az. dexteroporum* [16–19]. In addition, one species of the closely related genus *Amphidoma* (which, together with *Azadinium*, are included in the family Amphidomataceae), *Am. languida* also produces AZA [13, 19]. However, the currently known diversity of Amphidomataceae is much larger with 23 species in the family described to date, and most of those species were assigned as non-toxigenic based on cultured strains [20–22]. One main reason for the late discovery of these tiny unicellular organisms (most are ~ 10–15 µm in cell length) is their inconspicuousness, which challenges their detection and morphological description using traditional light microscopy. The recent development of molecular-based methods increasingly supports the detection of such inconspicuous organisms, predominantly in the context of harmful microalgae [23–26]. Particularly, quantitative polymerase chain reaction (qPCR) methodology gained momentum due to its high specificity and sensitivity [27]. By including standard curves of target DNA, the sensitive, qualitative detection is complemented with quantitative estimations of the target organism. Currently, the detection and quantification of toxic microalgae via qPCR has become a standard procedure in many studies and monitoring programs [28–30].

PCR-based assays have also been designed for Amphidomataceae. Toebe et al. [31] and Wietkamp et al. [32] developed specific TaqMan assays for three toxigenic species within this family (*Az. spinosum*, *Az. poporum* and *Am. languida*), and Smith et al. [33] designed a SYBR Green assay targeting the DNA of all amphidomatacean species. These qPCR assays have been used to detect Amphidomataceae in field samples, for example on research surveys along the Danish coastline [34], the Norwegian coast [35], the Irish coastline [32], the Argentine shelf [36] and in western Greenland [22], or are in use for the regular monitoring of *Az. spinosum* and *Az. poporum* in Irish coastal waters [37]. In 2018, the Marine Institute (Galway/Ireland) became the first laboratory internationally to validate and accredit the method for *Az. spinosum* detection in a routine monitoring programme to ISO 17025 standards. In these studies, the molecular qPCR assays assisted investigations on toxigenic Amphidomataceae and indicated their presence at many sites with low cell densities, when microscopy or LC-MS/MS were hardly able to detect the species or their respective toxins.

However, qPCR technology has some limitations. Obviously, a specific assay can only be designed if target sequences and reference target DNA (e.g., from cultured strains) is available. Moreover, new strains of similar or closely related species need to be included in updated specificity tests to avoid false positive results. The respective assay also requires recurrent validation by applying it to new strains of the target species. The latter mainly refers to a reliable DNA-based quantification of target cells, because different strains may vary in their ribosomal DNA (rDNA) copy number, which would strongly bias the quantification performance [38–40]. The reliable quantification of target DNA by qPCR within field samples is one of the most crucial factors, because (as already described) even the "gold standard"—microscopy—has drawbacks when it comes to inconspicuous organisms and the validation of the qPCR assay within the field matrix becomes challenging. Therefore, the combination of analytical methods with individual advantages has become a popular approach in studies on harmful microalgae to compensate limitations of single techniques [41–45].

Three of the four known AZA producing species (i.e., *Az. spinosum*, *Az. poporum* and *Am. languida*) are present in the North Sea. *Azadinium spinosum* was first isolated and described from the east coast of Scotland [15], and further strains were established originating from Ireland, Denmark and the Shetland Islands. All these strains share the same toxin profile (AZA-1, -2 and -33) and almost identical ITS sequences [46, 47]. Assigned as Ribotype A, these strains are clearly delimited from Norwegian strains with a toxin profile of mainly AZA-11 and -51, which form a different ribotype, Ribotype B [35]. Recently, Tillmann et al. [36] found evidence for a third ribotype (Ribotype C), consisting of *Az. spinosum* strains collected at the Argentinean Shelf in 2015, where no AZA were detected.

The second toxigenic species, *Az. poporum* was first described from the Danish North Sea coast [16], but is now known to occur in numerous regions around the globe [48] showing different toxin profiles and DNA sequences, with several ribotypes also identified [20, 49, 50].

The first strain of *Am. languida* was isolated from Bantry Bay at the Irish south-west coast in 2009 [51], but there is evidence that this species is widely distributed in the North Atlantic and North Sea [13, 34, 35]. Although no known AZA were found initially in the first *Am. languida* strain, two newly described AZA (AZA-38 and -39) were subsequently assigned as the dominant toxins of this species [19]. However, a different toxin profile was observed in an *Am. languida* strain from the Andalusian coast of Spain in 2017, consisting of AZA-2 and -43 [13], and AZA-52 and -53 have been detected in a few *Am. languida* strains from the Norwegian coast [35].

Although (toxigenic) Amphidomataceae are present in the North Sea and adjacent Atlantic areas, little is known about their spatial and temporal distribution. The aim of this study was to gain detailed data about the distribution and abundance of toxigenic Amphidomataceae and

AZA in the North Sea and in Irish coastal waters, and to provide information on the AZP risk in these regions. Therefore, in summer 2018, a research survey in the North Sea, the Celtic Sea and Irish coastal waters was undertaken. The vessel was equipped with light microscopes, a triple-quadrupole LC-MS/MS and a qPCR instrument for on-board visual-, chemical- and molecular-based detection and quantification of toxigenic Amphidomataceae and their toxins.

## Materials and methods

### Field campaign

The necessary field permits for this study complying with all relevant regulations were issued from The Netherlands (Rijksdienst voor Ondernemend Nederland), Belgium (Royaume de Belgique, Service public fédéral Affaires étrangéres), France (Ministére des Affaires Étrangéres et du Développment International), Ireland (Department of Foreign Affairs and Trade), and the United Kingdom (Foreign & Commonwealth Office) via the German Embassy and are available upon request.

Data were collected during the survey (HE-516) on-board *RV Heincke* between 17[th] July and 15[th] August, 2018. Starting in the German Bight, the vessel piloted through the English Channel, northwards towards the Irish south coast and then on a clockwise trajectory around the southwest, west and northwest coasts of Ireland, across to the Outer Hebrides and south of Orkney Islands in Scotland and returned to the German Bight again via a long southeastward transect through the North Sea. In total, 75 stations were sampled (station number and positions are listed in S2 Table), including six defined station transects (Fig 1). CTD profiles were recorded at each station using a Seabird 'sbe911+' CTD (Sea-Bird Electronics, Inc. Seattle, USA) with a sampling rosette (on-board device). A chlorophyll fluorescence sensor (Eco AFL-FL SN1365, Sea-Bird Electronics, former WET labs, Bellevue, USA) was attached to the CTD and used for the detection of chlorophyll maxima in the water column. CTD data were stored and processed using Seasave V 7.23.2. Temperature (potential T in ˚C) was calculated according ITS-90 [52]. Water samples were collected with Niskin bottles attached to the CTD

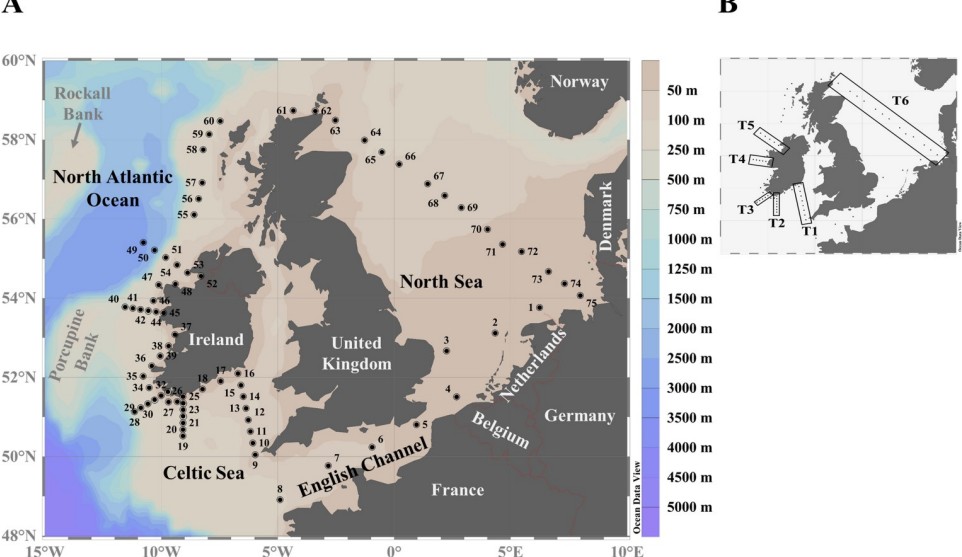

**Fig 1.** A: Study area and sampling locations during the survey HE-516. B: Transects (T1 –T6) referred to in the text are indicated by black rectangles.

from discrete depths during the upward casts. CTD data sets are available at Pangaea [53]. For field data correlations as well as visualization (S1 Fig), CTD profile data for temperature, salinity and fluorescence were averaged from the surface down to the maximum sampling depth.

At each station, plankton samples were collected with 10 L Niskin bottles at 3 m, 10 m and the deep-chlorophyll-maximum (DCM) layer. Five litres of seawater from each depth were filtered through a 20 μm mesh-size Nitex sieve, and the three depth samples were subsequently pooled and well mixed.

**On-board microscopy.** A defined volume (between 0.5 and 1 L) of the pooled water sample was gently concentrated by gravity filtration using a 3 μm Whatman polycarbonate filter (47 mm diameter, GE Healthcare, Little Chalfont, UK). The living plankton concentrate was collected in a flat (10 mm) 5 mL Utermöhl chamber and examined using an inverted microscope (Axiovert 200M, Zeiss, Göttingen, Germany). Cells were preliminarily identified as *Azadinium* and/or *Amphidoma* at high magnification (640 x) based on general cell size and shape, on the presence of a theca, and on the presence of a distinctly pointed apex. Cells of interest were photographed with a digital camera (Axiocam MRc5, Zeiss).

At 45 selected stations, the concentrated live samples were also used for a semi-quantitative estimation of Amphidomataceae cell densities. After at least one hour of sedimentation, the total number of amphidomatacean cells (without further differentiation into species or subgroups) was determined for a defined subarea of the chamber using 640 x and/or 1,000 x magnification. Assuming all cells of interest had settled to the bottom and taking the subarea/total chamber area ratio and the concentrated water volume into account, the Amphidomataceae cell abundance per L was calculated. The limit of detection of this counting procedure ranged (depending on the counted subarea and the sample volume) from ~ 20 to 50 cells $L^{-1}$.

**DNA sampling.** For DNA analysis, between 0.8–6.0 L (depending on the particle content) of the pooled water sample was filtered under gentle vacuum (< 200 mbar) through 3 μm pore-size polycarbonate filters (Merck KGaA, Darmstadt, Germany). For six randomly selected stations (19, 25, 32, 44, 64 and 71), the samples of the three different depths were not pooled, but analyzed separately. The filters containing the 3 μm phytoplankton fraction were placed inside a 50 mL plastic centrifuge tube (Sarstedt, Nümbrecht, Germany), extended along its inner side wall and vortexed with 30 mL of the 3 μm pre-filtered seawater for 1 min to resuspend the filtered particulate matter. The filter was removed and the 50 mL tube was then centrifuged (5415R, Eppendorf, Hamburg, Germany) at 3,220 × g for 15 min. The supernatant was discarded and the remaining cell pellet was subsequently collected in bead tubes together with 500 μL of the SL1 lysis buffer, both provided by the NucleoSpin Soil DNA extraction kit (Macherey & Nagel, Düren, Germany). The DNA was extracted immediately on-board according to the DNA kit manufacturer´s instructions with a slight variation. Instead of vortexing, the bead tubes were shaken in a cell disrupter (FastPrep FP120, Thermo-Savant, Illkirch, France) for 45 s initially and then for another 30 s, both run times at a speed of 4 $ms^{-1}$. DNA elution was performed using 2 x 50 μL of the provided elution buffer (to a final elution volume of 100 μL) to maximize the overall DNA yield. The DNA was stored at -20˚C until further processing.

**AZA sampling.** For AZA analysis, between 0.8–6.0 L (depending on the particle content) of the pooled water sample were filtered as described above. For the six selected stations (19, 25, 32, 44, 64 and 71), samples from all three depths were processed separately for AZA toxin analysis. Each filter was placed into a 50 mL centrifuge tube and extracted by a series of 0.5–1 mL methanol washes until complete filter discoloration. The extracts were subsequently transferred to a spin-filter (0.45 μm pore size, Millipore Ultrafree, Eschborn, Germany) and centrifuged (5415R, Eppendorf) for 30 s at 800 × g. The filtrate was then transferred to autosampler vials and immediately analyzed by LC-MS/MS.

## Sample processing

**qPCR.** Real-time qPCR with species-specific assays for *Az. spinosum*, *Az. poporum* and *Am. languida* was performed daily on-board as described in Toebe et al. [31] and Wietkamp et al. [32] using a Roche LightCycler 96 (Roche, Penzberg, Germany). All reactions were carried out in duplicate. Prepared positive Extraction Process Controls (EPC), which contained the DNA of each target species, as well as negative EPCs (0.4 μm filtered, sterilized seawater) were included during all PCR runs. The positive controls contained $10^3$ cells of each target species and the DNA of these cell pellets was extracted daily together with the field samples. This was to account for potential extraction efficiency variability between separate DNA extraction procedures, visible by comparing the $C_T$ (Cycle Threshold) values of the EPC samples between qPCR runs.

The final determination of cell densities of the three species for each sample (including samples of the spike experiment) was performed in technical triplicate after the survey at the Alfred-Wegener-Institute (Bremerhaven, Germany) according to procedures described previously [32]. $C_T$ values of the three technical replicates per sample were averaged and used for cell abundance calculation. The limit of quantification (LOQ) and the limit of detection (LOD) for these qPCR analyses were defined following Forootan et al. [54]. The LOQ referred to the lowest standard curve concentration, for which all three replicates showed amplification and are within the 95% confidence interval. The LOD was referred to the lowest standard curve concentration, for which all three replicates showed amplification but values are outside the 95% confidence interval of the standard curve. For the standard curves of all three qPCR assays on the field samples, the resolution of dilutions applied did not allow differentiation between LOD and LOQ, which were both 0.1 pg μL$^{-1}$.

**Evaluation of qPCR performance.** *qPCR assay specificity*. DNA of recently discovered target and non-target species and/or strains from Norway [35] and Denmark [32] was applied to the current qPCR assays on *Az. spinosum* and *Az. poporum*, to check whether the assays are still species-specific. Cell isolates of the respective species/strains were cultured and collected as described in Tillmann et al. [36]. For each sample, the DNA was extracted from cell pellets according to the procedures for the field samples. and normalized to a concentration of 1 ng μL$^{-1}$ for comparability of the amplification performance. Each sample was analyzed in triplicate in each of the qPCR assays. The specificity was evaluated by comparing the respective mean $C_T$ values to known target strains (UTH-D4, *Az. poporum* and 3D9, *Az. spinosum*; Table 1), which were used to design the original assay [31].

*Spike experiments*. To evaluate the qPCR assay performance in the field and to account for potential inhibition effects on the qPCR assay, known cell numbers of living cells of *Am. languida* (strain Z-LF-09-C9), *Az. spinosum* (strain 3D9) and *Az. poporum* (strain UTH-D4) were spiked on-board into a natural seawater matrix, taken at 20 m depth at station 3. Known target cell abundances were adjusted with the help of microscopic counts (magnification of 200 x) of 0.5 mL culture subsamples, settled at the bottom of counting chambers. The absence of the three species in the seawater matrix was confirmed by light microscopy and qPCR. In four replicates, 1 L seawater matrix each was spiked with three different total cell quantities ($10^2$, $10^3$ or $10^4$ cells) of all species, so that all three species were present. Negative controls, without added target cells, were also prepared. The spiked seawater sample was subsequently filtered through 3 μm filters, as described for the field samples, and then stored at -20°C until further processing. DNA of the spiked samples was extracted and species-specific qPCR assays were applied as described for the field samples.

**LC-MS/MS.** Water was deionized and purified (Milli-Q, Millipore, Eschborn, Germany) to 18 MΩ cm$^{-1}$ or higher quality. Formic acid (90%, p.a.), acetic acid (96%, p.a.) and

**Table 1. Strains used for specificity tests, including respective results of the qPCR assays.** n.a.: not analysed; ND: not detected.

| Species | Strain | Toxin profile | Ribotype | Reference | Result of the *Az. poporum* assay | Result of the *Az. spinosum* assay |
|---|---|---|---|---|---|---|
| *Am. languida* | N-01-01 | AZA-38, -39 | n.a. | [35] | ND | ND |
| *Am. languida* | N-01-02 | AZA-38, -39 | n.a. | [35] | ND | ND |
| *Am. languida* | N-40-03 | AZA-39, -52 | n.a. | [35] | ND | ND |
| *Az. dalianense* | N-12-04 | ND | B | [35] | ND | ND |
| *Az. dalianense* | N-38-02 | ND | A | [35] | ND | ND |
| *Az. obesum* | N-41-01 | ND | n.a. | [35] | ND | ND |
| *Az. polongum* | N-47-01 | ND | n.a. | [35] | ND | ND |
| **Az. poporum** | LF-14-E12 | ND | A | [55] | $C_T = 15.4$ | ND |
| **Az. poporum** | N-39-03 | AZA-37 | A | [35] | $C_T = 15.5$ | ND |
| **Az. poporum** | N-39-13 | AZA-37 | A | [35] | $C_T = 15.6$ | ND |
| **Az. poporum** | UTH-D4 | AZA-37 | A | [56] | $C_T = 15.3$ | ND |
| **Az. spinosum** | 3D9 | AZA-1, -2, -33 | A | [15] | ND | $C_T = 19.3$ |
| **Az. spinosum** | N-04-01 | AZA-1, -2, -33 | A | [35] | ND | $C_T = 19.4$ |
| **Az. spinosum** | N-04-04 | AZA-11, -51 | B | [35] | ND | $C_T = 25.7$ |
| **Az. spinosum** | N-05-01 | AZA-11, -51 | B | [35] | ND | $C_T = 26.8$ |
| **Az. spinosum** | N-16-02 | AZA-11, -51 | B | [35] | ND | $C_T = 25.2$ |
| *Az. trinitatum* | N-39-04 | ND | n.a. | [35] | ND | ND |

ammonium formate (98%, p.a.) were from Merck (Darmstadt, Germany). The solvents, methanol and acetonitrile, were high performance liquid chromatography (HPLC) grade (Merck, Darmstadt, Germany). Selected Reaction Monitoring (SRM) measurements were performed on-board monitoring for a wide array of AZA. The analytical system consisted of a API 4000 Q Trap triple quadrupole mass spectrometer equipped with a TurboSpray® interface (Sciex, Darmstadt, Germany) coupled to a model 1100 LC (Agilent, Waldbronn, Germany). The LC equipment included a solvent reservoir, in-line degasser (G1379A), binary pump (G1311A), refrigerated autosampler (G1329A/G1330B), and temperature-controlled column oven (G1316A).

Separation of AZA (5 µL sample injection volume) was performed by reverse-phase chromatography on a C8 phase. The analytical column ($50 \times 2$ mm) was packed with 3 µm Hypersil BDS 120 Å (Phenomenex, Aschaffenburg, Germany) and maintained at 20°C. The flow rate was 0.2 mL min$^{-1}$ and gradient elution was performed with two eluants, wherein eluant A was water and B was acetonitrile/water (95:5 v/v), and both contained 2 mM ammonium formate and 50 mM formic acid. A gradient elution was employed, starting with 30% B, rising to 100% B over 8 mins, held for 10 min, then decreased to 30% B over 3 min and held for 8 min to equilibrate the system. The profile of AZA was determined in one period (0–18 min) with curtain gas: 10 psi, CAD: medium, ion spray voltage: 5500 V, ambient temperature; nebulizer gas at 10 psi, auxiliary gas was off, the interface heater was on, the declustering potential at 100 V, the entrance potential at 10 V, and the exit potential at 30 V. The SRM experiments were carried out in positive ion mode by selecting the transitions shown in S1 Table. AZA were calibrated against an external standard solution of AZA-1 (certified reference material programme of the IMB-NRC, Halifax, Canada) and expressed as AZA-1 equivalents.

LODs for all congeners were defined as a signal-to-noise ratio = 3, evaluated individually for each sample.

**Statistics.** Statistics were computed using the open source program "R", version 3.4.3 [57]. Correlations between microscopic, qPCR, AZA and environmental data, as well as significance tests were performed with the implemented "pairs" and "corr test" functions (Pearson's

product-moment correlation test), respectively. The presented maps in this study were generated using the program "Ocean Data View" (ODV), version 5.1.0 [58].

# Results

## Hydrography and chemistry

Mean temperature (S1A Fig ) of the upper layer was highest in the German Bight and the North Sea entrance of the English Channel, reaching 18.5˚C at Helgoland (station 75) and 18.9˚C at station 4. Intermediate values (~ 15.5˚C) were observed at the Irish coast and the central North Sea stations, whereas low mean temperatures (< 13.5˚C) were measured along the Scottish coast, at northern North Sea stations, and at the western entrance of the English Channel (station 8). Salinity (S1B Fig) revealed higher values in the upper layer ($\geq$ 35) at stations off the coastline compared to coastal stations, with maxima of 35 in the central North Sea (stations 67–69) and 35.4 at Irish outer areas (stations 28, 40 and 49). Lower salinity values, down to 32.8, were obtained mainly in the German Bight (stations 1, 73, 74 and 75). Fluorescence values (S1C Fig) indicated higher densities of phototrophic organisms along the Irish west coast (up to 5.6 AU). A minimum fluorescence (0.4 AU) was measured at station 20, which was located on a transect at the Irish south coast. Mean fluorescence values in the upper layer in the range from 2 to 4 AU were observed in the German Bight (stations 1, 2, 71, 72, 74 and 75), Irish waters (stations 33, 34, 38, 39 and 47) and two stations (10 and 11) close to the coast of south west Wales.

Depth profiles for transect stations (Fig 1B) are presented in S2 Fig. For all transects there was a thermocline between the first 20 to 40 meters and a deeper water layer (S2A Fig), with stronger thermoclines occurring in transects T1, T2 and T6, where temperature differences of up to 10˚C were recorded. Salinity (S2B Fig) increased with distance from the Irish coastline. Higher salinity at the respective stations were not limited to the surface but were measured over the whole downcast of the CTD up to 100 m depth where maximum values exceeded salinity levels of 35.4. There were generally lower salinities along the North Sea transect, with minima ($\leq$34.6) recorded in the German Bight (stations 71–75). Fluorescence measurements (S2C Fig) revealed values mostly in the range from 0 to 4 AU. Values higher than 4 AU were infrequently detected and limited to certain water layers (e.g., station 10, 10–20 m depth). Peak values were found at station 11 (> 10 AU), 71 > 9 AU) and 72 (> 13 AU) as a deep (30–40 m) chlorophyll layer.

## Live sample records of Amphidomataceae

On-board microscopy using live samples revealed the presence of various species of Amphidomataceae. Species identification using light microscopy (LM) for the majority of species in this family is difficult and would require confirmation by scanning electron microscopy (SEM), but the detailed microscopical observation revealed the presence of at least nine different amphidomatacean species for the study area, including *Az.* cf. *zhuanum*, *Am. languida*, *Az. caudatum* (both varieties *Az. caudatum* var. *margalefii* and *Az. caudatum var. caudatum*), *Az. spinosum*, and *Az. obesum* (Fig 2). For a number of selected stations covering the western coast of Ireland, the eastern seaboard of Scotland, and the whole North Sea transect (S2 Table), the abundance of total amphidomatacean cells was estimated using a semi-quantitative LM method. Densities of all Amphidomataceae-like cells were estimated to range from "undetected" (i.e., below the detection limit of ~ 20–50 cells $L^{-1}$) to a maximum of 2.7 x $10^5$ cells $L^{-1}$ in the central North Sea. Highest density estimates for the Irish west coast were 1.8 x $10^4$ cells $L^{-1}$ (S2 Table).

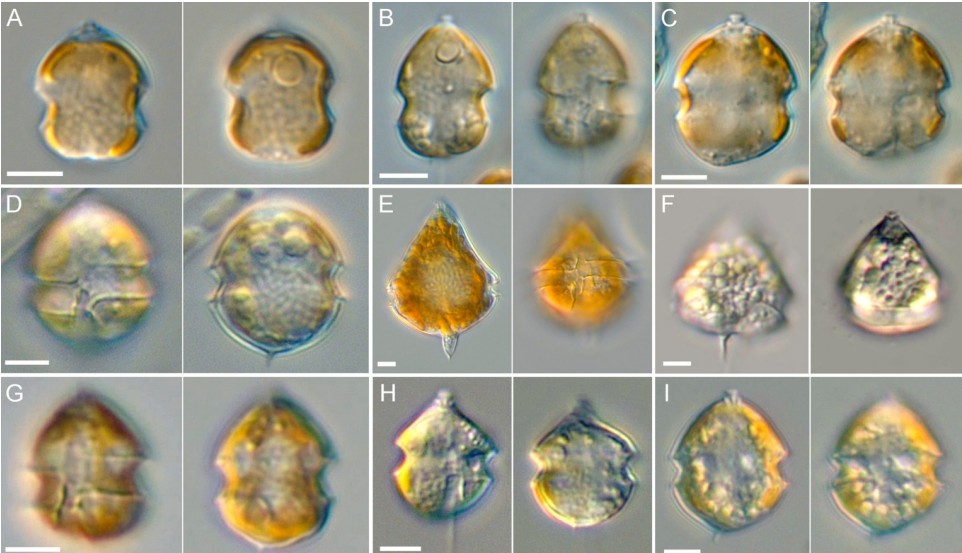

**Fig 2. Diversity of Amphidomataceae as recorded during HE-516 by live on-board light microcopy.** A: *Amphidoma languida*; B: *Azadinium spinosum*; C: *Az. obesum*; D: *Az.* cf. *zhuanum*; E: *Az. caudatum* var. *caudatum*; F: *Az. caudatum* var. *margalefii*; G: *Az.* spec. 1; H: *Az.* spec. 2; I: *Az.* spec. 3. Scale bars = 2 μm.

## qPCR assay performance

**Specificity tests on qPCR assays.** Application of several recently isolated target and non-target species/strains to the current *Az. spinosum* and *Az. poporum* qPCR assay revealed no amplification of non-target species DNA in both assays (Table 1). Newly isolated strains of *Az. poporum* from Norway (N-39-03 and N-39-13; Ribotype A1) revealed the same amplification efficiency ($C_T$ = 15.5 and 15.6) as the reference strain UTH-D4 ($C_T$ = 15.3). Likewise, DNA of the newly obtained strain from the Danish coast (LF-14-E12; Ribotype A2) was amplified with the same efficiency ($C_T$ = 15.4).

The Norwegian strain of *Az. spinosum*, belonging to Ribotype A (N-04-01), showed a similar amplification efficiency ($C_T$ = 19.4) as the reference strain 3D9 ($C_T$ = 19.3; also Ribotype A), whereas the Norwegian strains assigned as Ribotype B (N-04-04, N-05-01 and N-16-02) revealed less efficiency ($C_T$ = 25.7, 26.8 and 25.2).

**Spike experiment.** The on-board qPCR assay performance was tested for the three species-specific qPCR assays targeting *Az. spinosum*, *Az. poporum* and *Am. languida* by performing spike experiments (Fig 3). The recovery rate for different cell numbers of *Az. spinosum* and *Am. languida* ranged from 89 to 93%, the rate for *Az. poporum* was higher with a range from 93 to 111%.

Negative controls did not show any detectable amplification. Taking the filtered water volume and the DNA extraction volume into account, the LOD and LOQ was 3 cells L$^{-1}$ for the three assays.

## qPCR analyses of field samples

All three investigated toxigenic species (*Az. spinosum*, *Az. poporum* and *Am. languida*) were detected by qPCR (Fig 4). Considering the DNA extraction volume and filtered water volume, the limit of detection for the qPCR of 0.1 pg target DNA μL$^{-1}$ for all three species corresponded to 1 to 3 cells L$^{-1}$ in the field samples.

*Azadinium spinosum* was found at almost all stations along the Irish south and west coast. This species was also detected at five stations (67–71) in the central North Sea, but was not

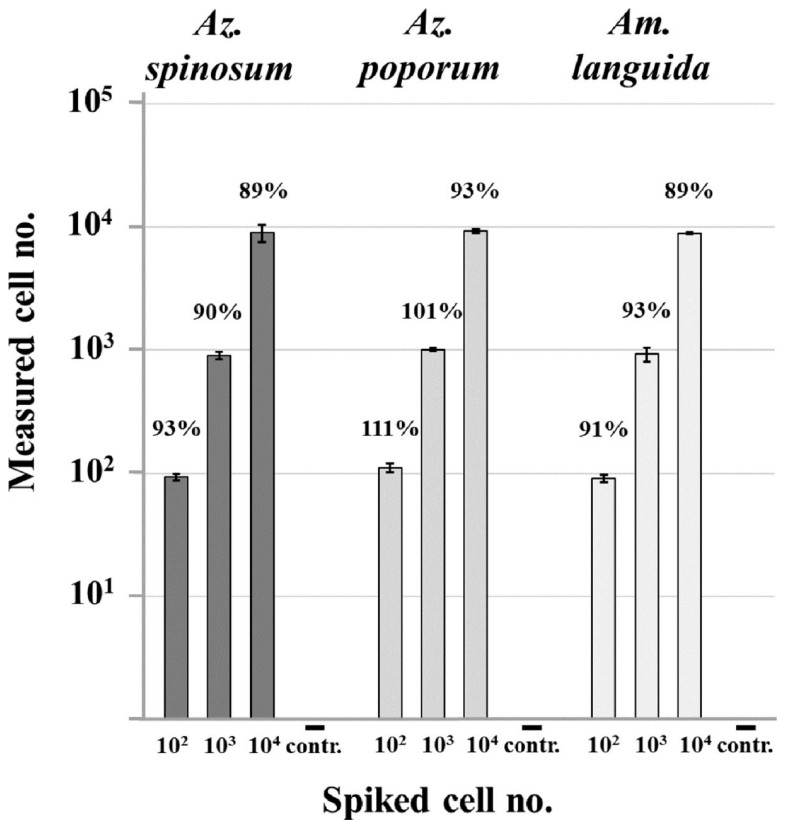

**Fig 3. Spike experiment: qPCR quantification of AZA-producer within environmental matrix.** Standard deviation is presented for each bar (n = 4). Numbers above the bars represent the percentage of recovery. Negative controls (-contr.) did not contain any target cells.

detected in the English Channel or in the Scottish coastal waters. *Azadinium poporum* was recorded at 18 stations. This species was mainly present along the southern Irish coast, at stations along the North Sea transect (stations 64, 65 and 69) and at two stations (stations 5 and 7) located in the English Channel, where *Az. poporum* was the sole toxigenic amphidomatacean species present. *Azadinium poporum* was not detected along the Irish west or Scottish coasts. *Amphidoma languida* was detected at majority of all stations (42 in total). It was present throughout the whole study area, with exceptions at stations inside the English Channel and a couple of stations off the Scottish coast. It was the only toxigenic amphidomatacean species detected in the German Bight and off the Outer Hebrides/Scotland. qPCR quantification revealed higher densities of *Az. spinosum* along the Irish south and west coasts, with peak cell densities at the coastal stations 31 (3.4 x $10^4$ cells $L^{-1}$), 45 (8.3 x $10^4$ cells $L^{-1}$), and 47 (2.6 x $10^4$ cells $L^{-1}$) (S2 Table). Relatively high abundances of 6.2 x $10^3$ cells $L^{-1}$ were also found in the central North Sea (station 71) and along a northward transect from the English Channel to the southern Irish coast (station 9–13). Overall, lower cell densities of the three targeted species were observed in the North Sea compared to the Irish stations.

*Amphidoma languida* was widely distributed, but in the Atlantic this species was lower in abundance than *Az. spinosum* (S2 Table). In the Atlantic area cell abundances of *Am. languida* rarely exceeded $10^3$ cells $L^{-1}$. In the North Sea, however, a maximum cell density of 1.2 x $10^5$ cells $L^{-1}$ was observed at the central station 71, which was the highest cell density of any of the three targeted species using qPCR during the survey.

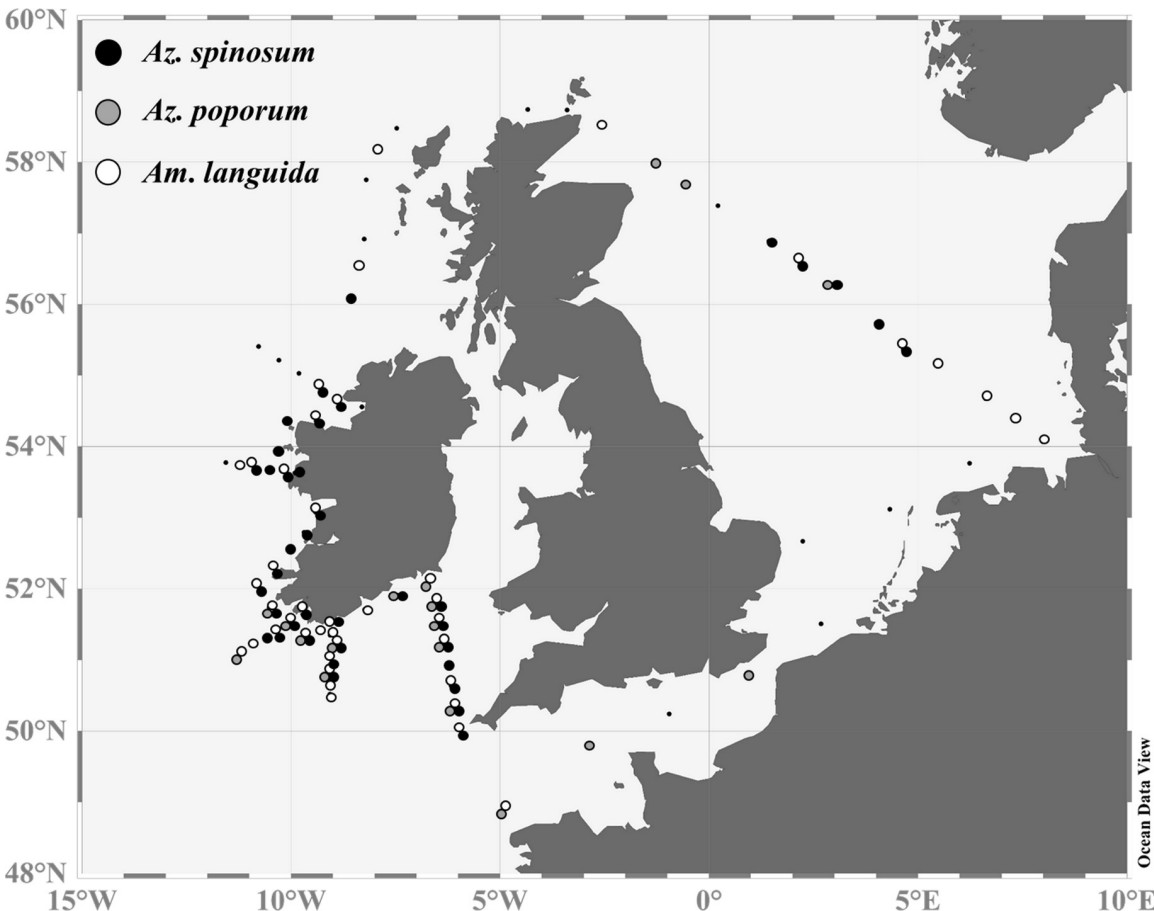

**Fig 4. qPCR qualitative results.** Positive hits for the species are indicated by black (*Az. spinosum*), grey (*Az. poporum*) and white (*Am. languida*) dots. Stations, where none of the three species were detected, are presented as smaller black dots.

Plotting species abundance along six station transects (Fig 1B) revealed that *Az. spinosum* was more abundant at the south-western and western transects (up to $8.3 \times 10^4$ cells $L^{-1}$ at station 45, T4) than along the southern stations, where a maximum of $2.4 \times 10^3$ cells $L^{-1}$ was found at station 12, T1 (Fig 5). The peak density on the North Sea transect (T6) was $6 \times 10^3$ cells $L^{-1}$ at station 71. In contrast, *Am. languida* (although generally in lower cell densities than *Az. spinosum*) was more abundant along the south-western Irish coast (up to $1.5 \times 10^3$ cells $L^{-1}$ at stations 26 and 35) compared to the western stations with a maximum of $1 \times 10^2$ cells $L^{-1}$ at station 42. Stations on the North Sea transect (T6) revealed higher abundance of *Am. languida* in the North Sea area compared to Irish waters, and especially at stations in the central North Sea and towards the German Bight (stations 71–75, with the peak density of $1.2 \times 10^5$ cells $L^{-1}$ at station 71) (Fig 5). In T1 and T2, the southern Ireland transects, cell densities of both species (*Az. spinosum* and *Am. languida*) were higher in the middle stations of each transect. However, for the south-western and western transects T3 –T5, highest cell densities were observed at the stations closer to shore (Fig 5).

## Chemical detection of AZA in field samples

In total, five AZA (AZA-1, -2, -33, -38 and -39) congeners were detected by LC-MS/MS within the 3–20 μm size plankton fraction. The individually determined LOD of single compounds (depending on the filtration and extraction volume) was in the range 5–50 pg $L^{-1}$. AZA-1 was

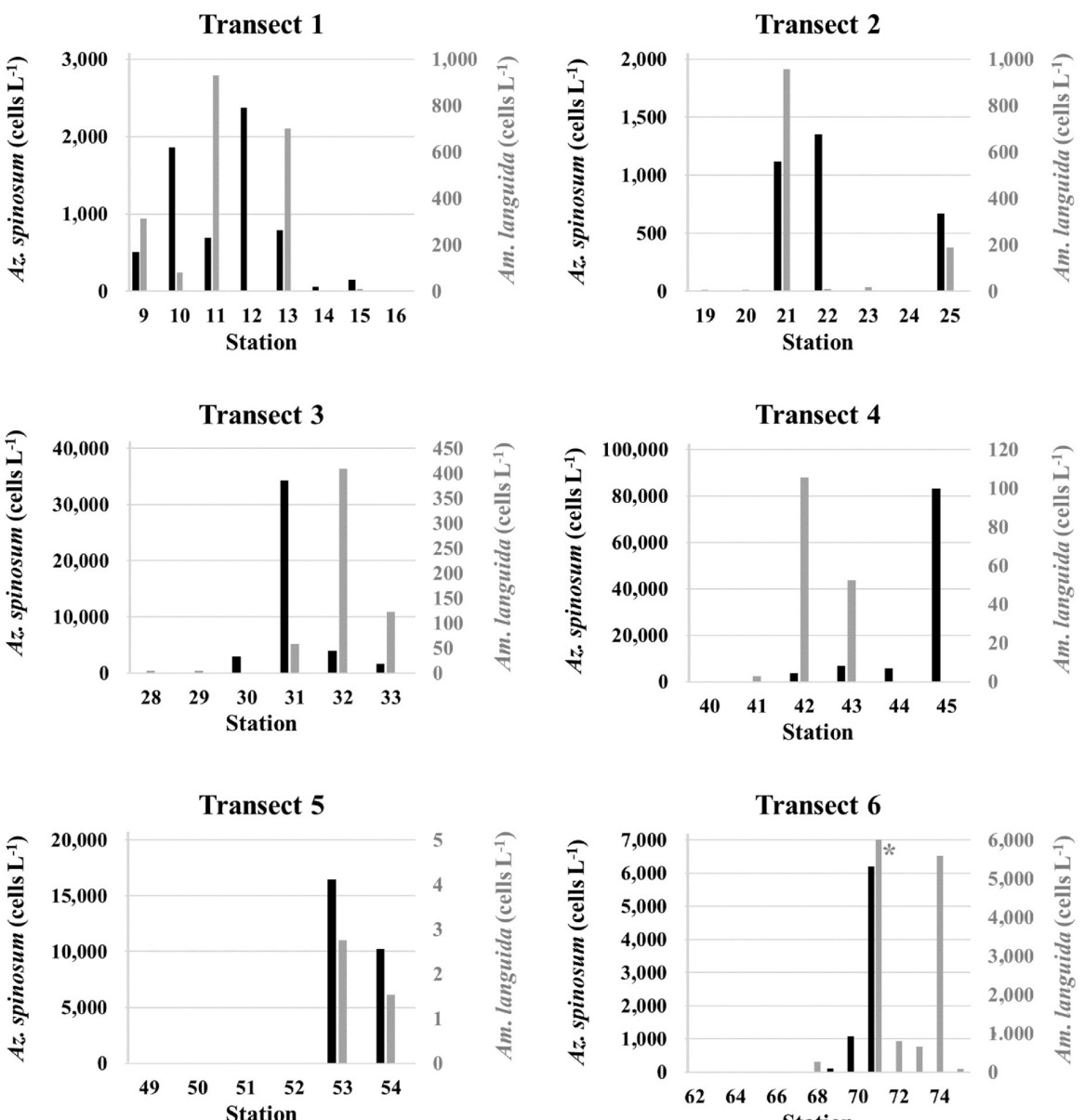

**Fig 5. Cell numbers (cells L⁻¹) estimated by qPCR for *Az. spinosum* (black bars) and *Am. languida* (grey bars) for the six transects** (T1 –T6) shown in Fig 1B. Cell number calculation based on mean $C_T$ values of three technical replicates. The peak cell density ($> 120,000$ cells L⁻¹) of *Am. languida* at station 71 is not shown, but indicated by the grey asterisk in T6.

detected at 31 stations, with a mean amount (samples with AZA-1 $\geq$ LOD) of 87 pg L⁻¹ and peak values of 676 and 745 pg L⁻¹ at the western Irish stations 44 and 45, respectively (S2 Table). At most stations, where AZA-1 was detected, AZA-2 and AZA-33 were also observed but generally at lower quantities compared to AZA-1 (mean AZA-2: 62 pg L⁻¹; mean AZA-33: 37 pg L⁻¹). Peak values with 325 pg L⁻¹ (AZA-2) and 204 pg L⁻¹ (AZA-33) were recorded at stations 44 and 45 (S2 Table). AZA-38 and -39 were less prevalent and only detected at two stations in the central North Sea. Rather high levels of AZA-38 (234 pg L⁻¹) and AZA-39 (384 pg L⁻¹) were measured at station 71, whereas levels of these two compounds at the neighboring station 72 were lower (S2 Table).

## Comparative method analysis of field data

There was no significant correlation between any of the measured environmental parameters (temperature, salinity and oxygen) with neither amphidomatacean abundances calculated by microscopy or qPCR, nor with AZA quantities measured by LC-MS/MS.

**Microscopy and qPCR.** Cell density (cells $L^{-1}$) estimates obtained by semi-quantitative live-microscopy counts (as total Amphidomataceae) and qPCR (sum of the three species-specific assays) revealed similar biogeographical patterns (Fig 6A and 6B and S2 Table). Both methods revealed high abundances along the Irish west coast, with peak cell estimates of 1.8 x $10^4$ cells $L^{-1}$ based on microscopy, and 8 x $10^4$ cells $L^{-1}$ based on qPCR at station 45. In the North Sea both methods identified maximum densities at station 71 where 2.8 x $10^5$ cells $L^{-1}$ (microscopy) and 1.3 x $10^5$ cells $L^{-1}$ (qPCR) were recorded.

Total Amphidomataceae (microscopy) and toxigenic species (qPCR) generally were in the same order of magnitude and were positively correlated (R = 0.89, Pearson's product-moment correlation test: p < 0.001) (Fig 7). The discrepancy between qPCR estimates and microscopic counts was generally higher for the North Sea stations than for the stations located along the Irish west coast.

**qPCR and AZA.** qPCR abundance and toxin data are compared under the assumption that AZA-1, -2 and -33 are produced by *Az. spinosum*, whereas AZA-38 and -39 are produced by *Am. languida*. Both species-specific qPCR data for *Az. spinosum* and *Am. languida*, and their respective toxins, revealed similar biogeographical patterns (Fig 6C–6F). qPCR cell density estimates of *Az. spinosum* were positively correlated (R = 0.76 to 0.85) to AZA-1, -2 and -33 quantities (Fig 8), and Pearson's product-moment correlation tests revealed highly significant correlations (p ≤ 0.001) between the four variables. A number of samples revealed positive qPCR hits for *Az. spinosum*, but no AZAs were detected. This was the case in eight stations for AZA-1, in 17 stations for AZA-2 and 19 stations for AZA-33 (Fig 8 and S2 Table). There was only one station (station 24) where AZA was detected (12 pg $L^{-1}$ AZA-1), but no qPCR amplification for *Az. spinosum* DNA was observed. Positive and highly significant correlations (R = 0.86 to 0.91; p ≤ 0.001) were found also between AZA-1, -2 and -33 levels.

AZA-38 and -39 were only detected at two stations (stations 71 and 72) and thus the correlation of cell number estimates of *Am. languida* by qPCR and AZA-38 and -39 were not calculated.

Field data estimates of AZAs and qPCR-based cell densities allowed calculation of AZA cell quotas. Mean cell quotas for *Az. spinosum* and AZA-1, -2 and -33 were 17.8 ± 9.9, 8.3 ± 11.0 and 3.0 ± 1.5 fg $cell^{-1}$, respectively (Fig 9). Maximum values for AZA-1, -2 and -33 were 46.4, 43.2 and 5.9 fg $cell^{-1}$, respectively. The respective AZA cell quotas of *Am. languida*, calculated for the two North Sea stations where AZA-38 and -39 were detected, revealed 1.9 and 3.1 fg $cell^{-1}$ for AZA-38 and -39 at station 71, and 101.0 and 92.3 fg $cell^{-1}$ for station 72.

The depth distribution of AZA and toxigenic Amphidomataceae, as estimated on six selected stations (19, 25, 32, 44, 64 and 71), was quite variable between stations. At station 25, 44, and 71 most cells were found at the Deep-Chlorophyll-Maximum (DCM) layer (note the log-scale in Fig 10). Station 71 showed the highest cell density for toxigenic *Am. languida*, but also relatively high amounts of *Az. spinosum*. The CTD depth profile revealed almost a constant salinity value between the surface and 40 m. A distinct thermocline at ~ 27 m depth was observed with a sudden decrease of the potential temperature from 18 to 6.5˚C. Around the thermocline an exceptional high phytoplankton biomass was present, as indicated by distinct peaks in fluorescence (~ 15 AU) and oxygen (~ 340 µmol $L^{-1}$, oxygen data can be found at [53]).

At station 19 there was a deep (~ 55 m) thermocline with a weak fluorescence signal and at station 64 there was a strong fluorescence signal at 23 m depth. At both stations, the few cells of toxigenic Amphidomataceae were recorded in the sub-surface samples only. Higher cell

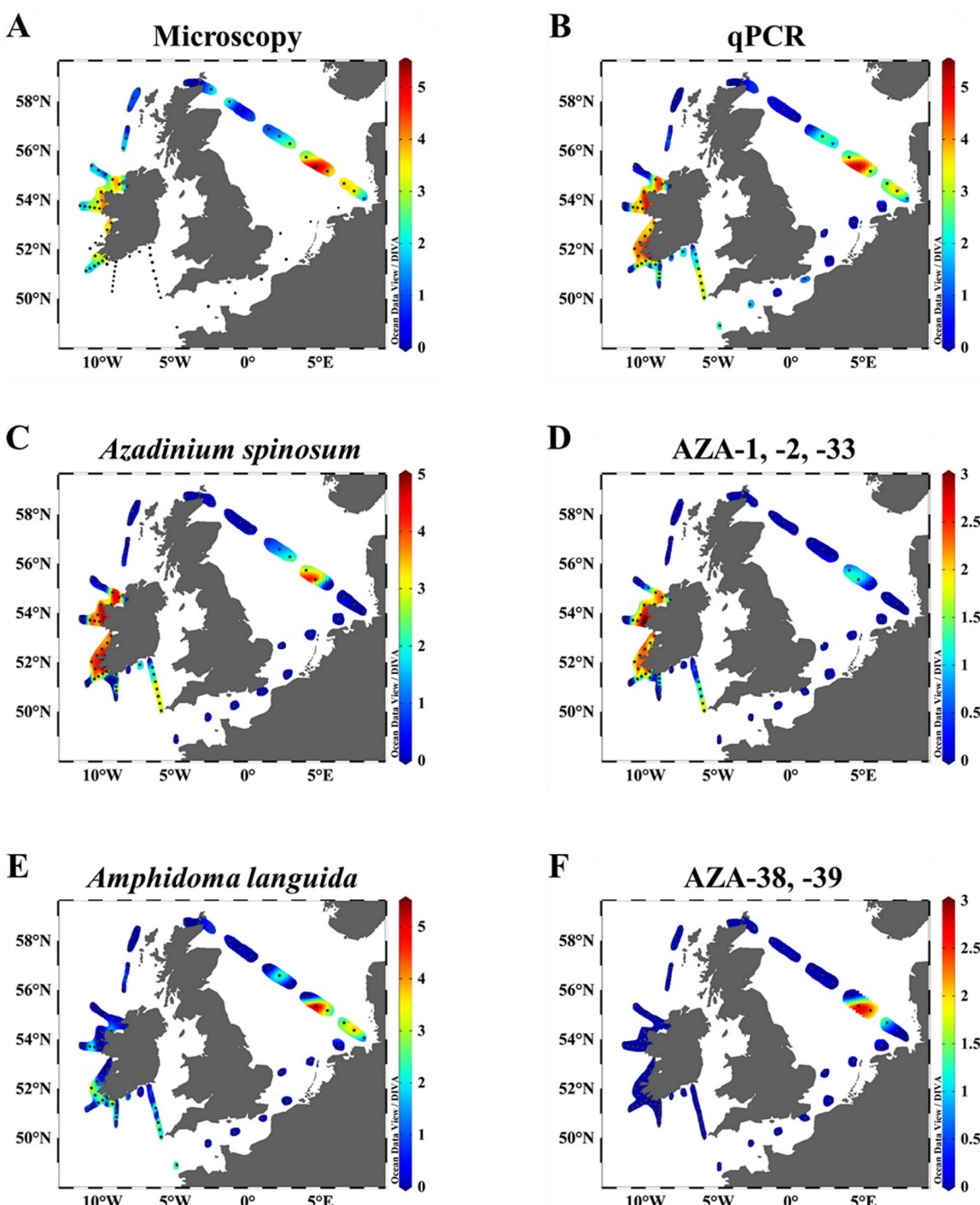

**Fig 6. Graphical visualization of general agreement between on-board light microscopy, qPCR, and chemical AZA quantification.**
Color code refers to the logarithmic scale ($\log_{10}$). A–B: Abundances of amphidomatacean cells $L^{-1}$ by (A) light microscopy and (B) qPCR estimations. C–D: Cells $L^{-1}$ of *Az. spinosum* estimated by qPCR (C) and AZA quantities (pg $L^{-1}$) of AZA-1, -2 and -33 (D) measured by LC-MS/MS. E–F: Cell numbers $L^{-1}$ of *Am. languida* estimated by qPCR (E) and AZA quantities (pg $L^{-1}$) of AZA-38 and AZA-39 (F) measured by LC-MS/MS.

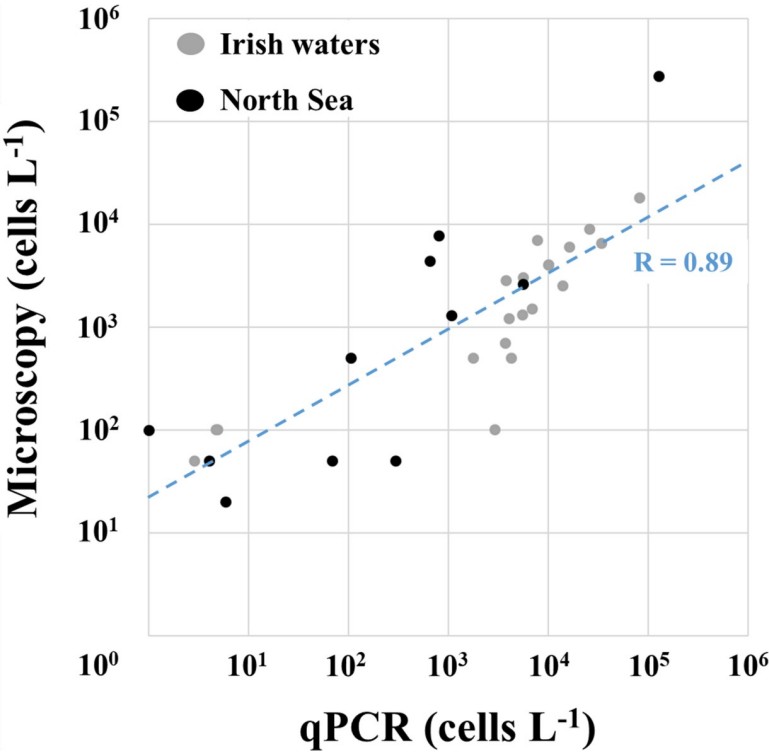

**Fig 7. Correlation between microscopic (x-axis) and qPCR (y-axis) calculated cell numbers L⁻¹.** Dots indicate the sample area of either Irish waters (grey) or the North Sea (black). "R" displays the respective correlation coefficient.

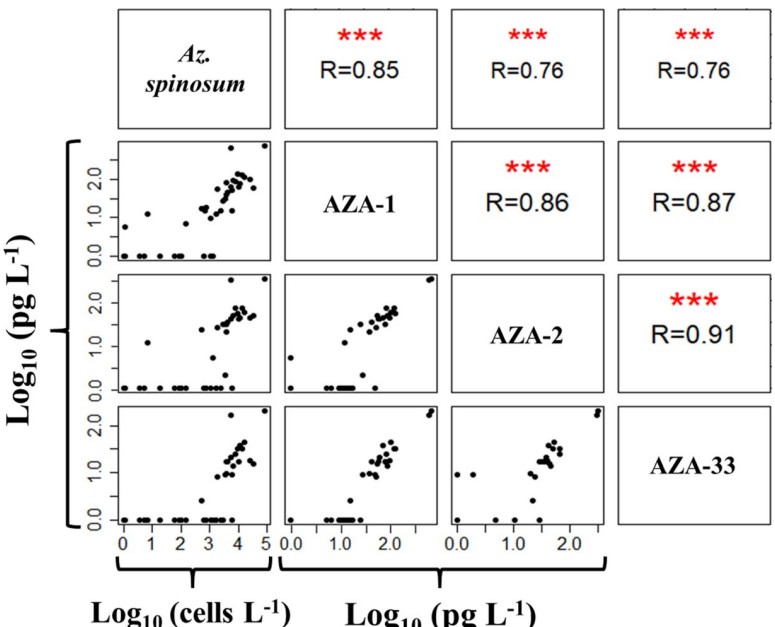

**Fig 8. Pearson correlation matrix of logarithmic qPCR counts (cells L⁻¹) and logarithmic AZA quantities (pg L⁻¹).** "R" displays the respective correlation coefficient. Significance levels of respective correlations (Pearson's product-moment correlation test) are indicated by red asterisks (* $p \leq 0.05$, ** $p \leq 0.01$, *** $p \leq 0.001$).

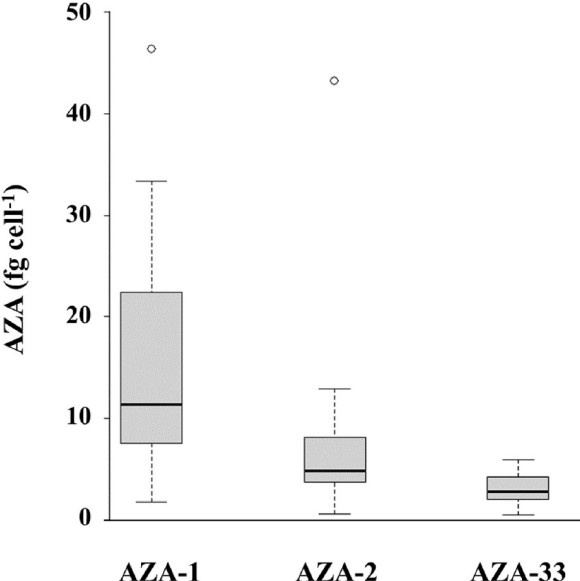

**Fig 9. AZA cell quota for *Az. spinosum* and AZA-1, -2 and -33 based on molecular (qPCR) and chemical (LC-MS/MS) field observations.**

densities of toxigenic species in the upper 10 m were also observed at station 32, where neither a thermocline nor a deep chlorophyll maximum was observed. The depth-distribution pattern of AZA mirrored the cell distribution, but for samples with lower cell density estimates (e.g., stations 19 and 64) no AZAs were detected.

## Discussion

### Amphidomataceae and AZA in the North Atlantic: Distribution and abundance

Microscopy, species-specific qPCR assays and LC-MS/MS revealed that Amphidomataceae, and especially all three North Atlantic toxigenic species, are widely distributed in Irish coastal waters, the Celtic Sea and the North Sea. All three species identified and quantified using qPCR revealed different abundances, with *Az. spinosum* and *Am. languida* being more abundant and widespread compared to *Az. poporum*.

**Amphidoma languida.** *Am. languida* was detected at most stations. Most notably, the highest amphidomatacean cell abundance, of more than $10^5$ cells $L^{-1}$ observed at station 71 in the North Sea transect, was mainly dominated by *Am. languida*. AZA-38 and -39, the two AZA-congeners known to be produced by North Atlantic strains of *Am. languida* [59], were also detected at these North Sea stations, with relatively high toxin levels of 234 pg $L^{-1}$ and 384 pg $L^{-1}$, respectively.

This peak of amphidomatacean abundance in the North Sea at station 71 was associated with a dense and deep (~ 30 m) chlorophyll patch extending from stations 71 to 72. While *Am. languida* was much more abundant at station 71 compared to 72, fluorescence was much higher at station 72, which may indicate that different plankton communities or different development stages were sampled. In any case, this subsurface bloom appeared close to the shelf break where water depth decreased from > 100 m to ~ 50–40 m (S2C Fig). This shelf area off the German Bight is generally regarded as a highly productive habitat [60, 61]. Summer subsurface chlorophyll peaks have been observed previously to be prominent in the stratified

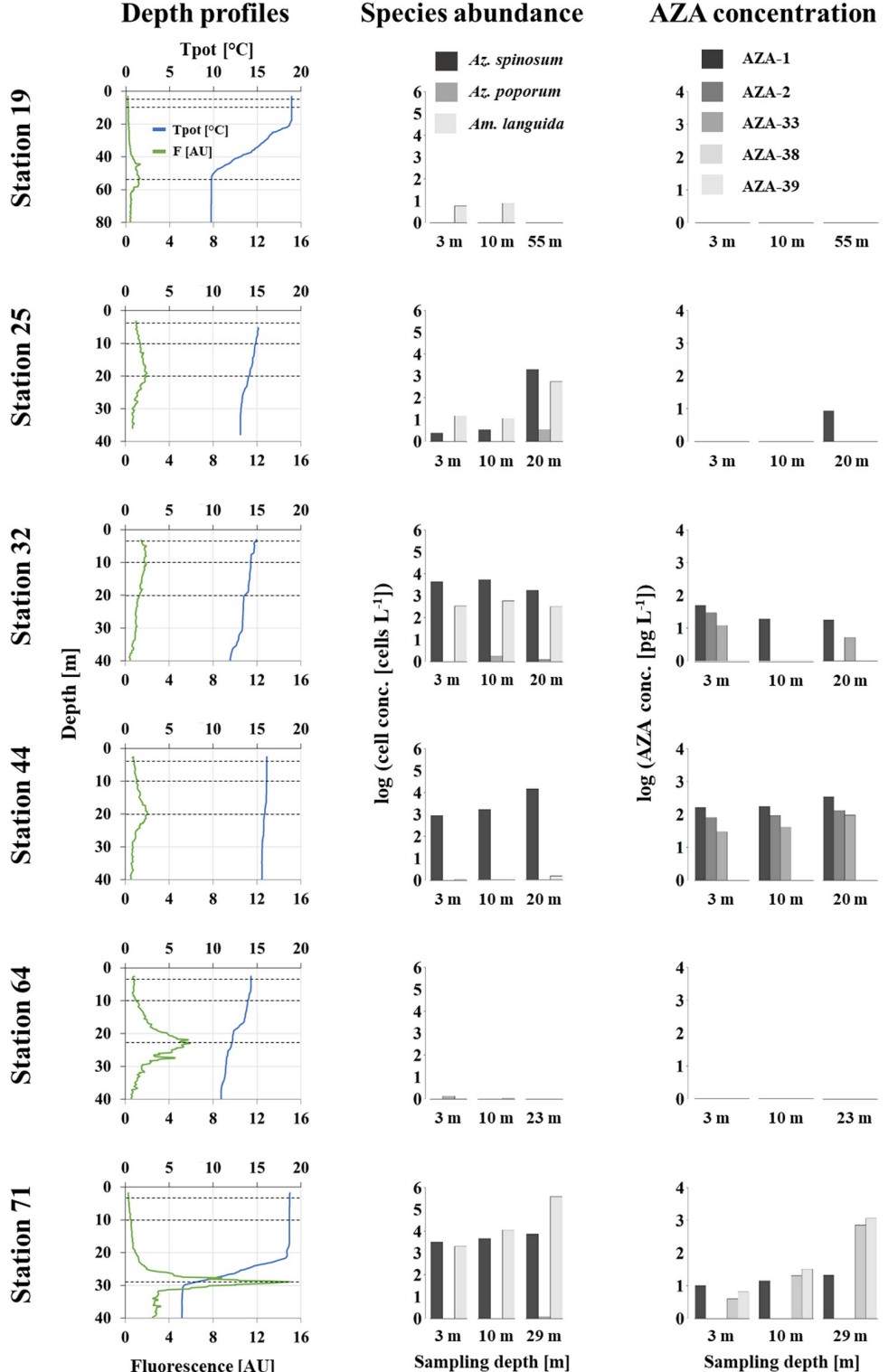

**Fig 10. Depth (m) profiles of potential temperature Tpot (°C), fluorescence (AU), AZA-producer abundances and respective toxin quantities for six selected stations.** Dashed lines in the profile plots indicate the depth for the respective samples.

regions of the North Sea, to be patchily distributed both in time and space, and to be dominated by dinoflagellates and not by diatoms [62]. For the formation of such blooms a 'tidal pumping' mechanism entailing the fortnightly sweep of the tidal mixing front from shallow to deep water and thereby injecting nutrient-rich bottom water into the pycnocline layer, has been discussed [63].

The maximum *Am. languida* density of this North Sea bloom of $>10^5$ cells L$^{-1}$ is the highest density of an amphidomatacean species yet reported from the North Atlantic and/or North Sea waters. Blooms of other Amphidomataceae in other geographical regions have been found at even higher abundances, with records up to $10^6$ (*Az. polongum*, Peru) or up to $10^7$ (*Az. luciferelloides*, Argentina) cells L$^{-1}$ [64, 65].

Based on the present findings, *Am. languida* appear to be the dominant AZA-producers in the central North Sea and the German Bight compared to *Az. spinosum* and/or *Az. poporum*, however, this survey is only a snapshot in time. In any case, a dominance of *Am. languida* over *Az. spinosum* for the North Sea matches with data from a previous field study on toxigenic Amphidomataceae in the North Sea [34]. *Amphidoma languida* was also widely present around the Irish coast. Application of the *Am. languida* specific qPCR assay on samples taken on a 2017 Irish coastline survey (CV17022) revealed overall cell abundances of up to ~ 1.5 x $10^3$ cells L$^{-1}$ with peak densities of up to 2.3 x $10^4$ cells L$^{-1}$ along the Irish coastline [32]. Together with the present results these data sets indicate a frequent occurrence and significant abundance of *Am. languida* at the Irish southern and south-western coastline and emphasize the potential risk for shellfish contamination with *Am. languida* specific AZA in these production areas. However, different to the high levels of AZA-38 and -39 found in the central North Sea, these two *Am. languida* specific AZA congeners were not detected around Ireland (Fig 6F and S2 Table), which might be explained by the limit of detection for the chemical method which may not be sensitive enough to detect the prevalent but low abundances. Nevertheless, with the emerging importance of the presence of *Am. languida* in the North Sea and the Irish Atlantic the routine surveillance of both *Am. languida* cells (applying the specific qPCR assay) and the presence of AZA-38 and -39 should be considered. Both are currently not regulated by the European Union (EU) in routine monitoring programs. Importantly, however, actual *in vivo* toxicity of AZA-38 and -39 is still unknown and has to be investigated.

**Azadinium spinosum.** *Azadinium spinosum* showed a distribution similar to *Am. languida* and both species co-occur in Irish coastal waters as well as in the North Sea. Peak cell densities and levels of AZA congeners typically produced by *Az. spinosum* (i.e., AZA-1, -2 and -33) were clearly concentrated at the Irish southwestern and western coastline (Fig 6C). This underlines the special threat of *Az. spinosum* to Irish aquaculture [11]. Previous laboratory growth studies revealed that *Az. spinosum* are able to cope with a wide range of environmental conditions. Temperature had the most significant impact on the growth of a North Sea strain and indicated higher growth rates at relatively high temperatures of 22˚C [66], however toxin production was significantly higher at lower temperatures [66, 67]. Therefore, water temperatures of 13 to 17˚C, as observed in this study, are likely to be suboptimal for rapid cell division and formation of extensive *Az. spinosum* blooms. On the other hand, the strong thermocline, as observed in the central North Sea at station 71 (Fig 10), had relatively high temperatures of ~ 18˚C and thus might have favored the high cell densities of *Az. spinosum* and *Am. languida*.

The presence of *Az. spinosum* cells in the North Atlantic area off the west coast of Ireland have been observed previously [11]. A transatlantic survey conducted in 2014 from Galway (west coast Ireland), revealed the presence of *Az. spinosum* cells at several stations along a transect through the North Atlantic area, up to the Bight Fracture zone, where the highest cell density observed was 1.3 x $10^4$ cell L$^{-1}$ at a station in the Porcupine bank and the second highest cell density was observed at 1 x $10^4$ cells L$^{-1}$ between the Edoras and Fangorn Banks past the

Rockall Bank [68]. Along the Irish coastline, prevalent abiotic parameters, like wind direction and prevailing winds, could allow for bloom formation by concentration of plankton inside bays, even under suboptimal growth conditions. This might be of special importance for the south-west coastal bays, which are open directly to these western-southwesterly winds. Dino-flagellates are associated with the thermocline and surface mixed layer brought into the bays by downwelling pulses [69, 70]. Data using satellite tracked drifting buoys on the north west European shelf [71] reveal that water mass circulation around the Celtic Sea follows a highly organized thermohaline circulation. This circulation advects water through the south and west of the region of St. George's Channel. This flow is directed south into the Celtic Sea and west along the southern Irish coast [72]. Furthermore, there is evidence that this flow extends around the south-western tip of Ireland [71–73]. Typical northeastward winds in the region probably play an important role in the wind driven advection of plankton into the bays of the south-west of Ireland [74]. This circulation pattern may explain why the higher cell densities were found at the inner stations of transects along the southwest and west coast of Ireland (T3, T5) and further offshore in southern Ireland transects (T1 and T2, Fig 5). Nevertheless, the impact of abiotic factors on the growth and bloom formation of toxigenic Amphidomataceae remains poorly known and additional studies should be conducted to address this knowledge gap.

qPCR records of *Az. spinosum* coincide with LC-MS/MS records of the AZA congeners produced by Irish and North Sea strains (i.e., AZA-1, -2 and -33). Both data sets were signifi-cantly correlated and allowed the calculation of the field population AZA cell quota. With a mean AZA-1 cell quota of 17.8 fg cell$^{-1}$ (Fig 9) the field measurements are in the same range as laboratory measurements of isolated strains, which typically vary between ~ 1 and 20 fg cell$^{-1}$ [11, 15, 36] and were higher (up to 200 fg cell$^{-1}$) when grown at 10˚ C [66]. The range of AZA cell quota as estimated here (e.g. AZA-1 cell quota ranging from 2–46 fg cell$^{-1}$, and AZA-38/ -39 cell quota ranging from 2–101 fg cell$^{-1}$) likely include methodological issues (e.g., chemical AZA analyses may include AZA accumulated in small protistan grazers). There may also be physiological reasons e.g., cell quotas of *Azadinium* spp. usually increase when growth is lim-ited/stagnant [67, 75]. The toxin profile of AZA-1, -2 and -33 is typical of Irish, Scottish and Shetland strains [46], which in phylogenetic trees form a well-supported ribotype (Ribotype A) [35]. Other co-occurring *Az. spinosum* strains from the Norwegian coast, however, cluster in another *Az. spinosum* ribotype (Ribotype B) and have a fundamentally different AZA profile consisting mainly of AZA-11 and -51 [35]. In the present study, only AZA-1, -2 and -33, but no AZA-11 or -51 were detected, suggesting that Ribotype B was not present, or perhaps in low abundance. First and yet unpublished characterization of multiple *Az. spinosum* strains established from the survey confirm that strains from Ireland are exclusively of Ribotype A, whereas the majority of *Az. spinosum* strains from the central North Sea are of Ribotype B.

**Azadinium poporum.** Based on various records around the world, *Az. poporum* seems to be the most widely distributed species of Amphidomataceae [48]. The species doubtlessly is also present in the North Sea and Irish waters but during the survey it was restricted to the southern Irish coast and the North Sea in very low abundances ($< 60$ cells L$^{-1}$) compared to the other toxigenic Amphidomataceae. Previous Irish coastline surveys conducted in 2012 and 2016 to 2019 have also shown low abundance or complete absence of *Az. poporum* in samples [68]. Notably, it was the only toxigenic amphidomatacean species detected in the English Channel. This area is known for its seasonal dynamics, spatial heterogeneity and inter-annual variability of harmful algae [76], so more data are needed to evaluate the importance of local weather conditions and current circulation pattern for the occurrence and abundance of Amphidomataceae in the English Channel.

**Vertical distribution of AZA producers.** Little is known about the depth distribution of AZA-producing species. Original data, as obtained here at three different depths at six stations, revealed variable patterns in species depth distribution. Highest densities of toxigenic Amphidomataceae at the deep-chlorophyll-layer in the North Sea indicate that surface water sampling and/or satellite observations are less well suited for the detection of such Amphidomatacean blooms. In any case, the limited depth-distribution data set presented here indicates that reducing the number of samples by pooling (= averaging) different depths seems to be a justified strategy to estimate the occurrence and abundance of AZA producers in the water column. While this method might in some cases shift cell densities per volume of filtered seawater below the "cell detection limit" of the LC-MS/MS, overall it is a cost and time effective sampling procedure.

**Critical evaluation of qPCR quantification.** The spike experiments for all three toxigenic species with an average recovery efficiency of ~ 94% of the target cells over a range of $10^2$ to $10^4$ target cells per L revealed a very robust assay design and reliability of the quantitative field data calculations. Specific qPCR is thus confirmed to be a valuable tool for the detection and enumeration of low abundant species [77]. In addition to this and previous Amphidomataceae qPCR spike-recovery experiments [32], a direct comparison of qPCR quantification with other independent quantification methods over a larger data set is desirable. Comparison with optical detection and quantification using target-specific, optical markers, for example by fluorescence *in situ* hybridization (FISH), seems to be the most straightforward strategy. Molecular FISH assays for toxigenic *Az. spinosum* and *Az. poporum* are available but labour-intensive [31]. More important, however, these assays were designed at a time when only three species and a limited number of strains were known. Extensive testing would be needed to exclude cross reactivity of these FISH assays with any of the currently known species, most of which are non-toxigenic amphidomatacean species.

Nevertheless, qPCR data can be compared to the semi-quantitative microscopy quantification of live amphidomatacean cells, even if the latter method captures all Amphidomataceae (including toxigenic and non-toxigenic species), whereas qPCR specifically quantifies the three toxigenic species only. With all these limitations in mind, there is generally a good agreement in cell abundances between both methods. However, some discrepancies between microscopy and qPCR analysis were observed for some stations. Higher cell estimates by live microscopy can easily be due to varying levels of non-toxigenic amphidomatacean species, which undoubtedly are present (Fig 2) but not included in the qPCR estimates. Moreover, lower *in vivo* rDNA copy numbers of the local target population than of the strain used to calibrate the qPCR assay can easily lead to a qPCR underestimation of the actual cell densities and vice versa [39]. Higher cell densities calculated by qPCR compared to microscopic counting might also be due to limitations of the live counting method, if not all cells present in the sedimentation chamber sink to the bottom and/or are identified correctly as members of Amphidomataceae.

qPCR-based cell abundances and AZA levels measured by LC-MS/MS in the field samples were also well correlated. This could be demonstrated adequately for *Az. spinosum* and its corresponding toxins AZA-1, -2 and -33. Out of 44 stations, where *Az. spinosum* was detected by qPCR, 35 stations also revealed its synthesized AZAs. The non-correlation of 20% was likely due to the higher detection limit of LC-MS/MS. The mean limit of detection for AZA-1 in the field samples was 13.2 pg L$^{-1}$. Given the calculated AZA cell quota of 17.8 fg cell$^{-1}$ (AZA-1 produced by *Az. spinosum*, this study), the "cell detection limit" corresponds to ~ 1 x $10^3$ cells L$^{-1}$. Such a "threshold" can also be seen from the depth profile samples (Fig 10): cell densities calculated by qPCR above a density of ~ 1 x $10^3$ cells L$^{-1}$ correspond to positive AZA measurements, whereas almost no AZA was recorded for qPCR cell abundance levels < 1 x $10^3$ cells L$^{-1}$ (S2

Table). Of all field samples, only three stations revealed AZA-1 signals although the qPCR estimated cell densities below 500 cells $L^{-1}$. Toxin accumulation in small protistan grazers and/or bound to decaying cells or detritus may be an explanation.

## Conclusion

Live cell microscopy, chemical AZA analysis and qPCR assays revealed that toxigenic Amphidomataceae and AZA were widely distributed in summer 2018 in the eastern North Atlantic. Quantification of three toxigenic amphidomatacean species was based on qPCR assays, whose performance was challenged using extended specificity testing and spike-recovery experiments. qPCR quantification data had a highly significant correlation with total amphidomatacean cell densities based on live-cell microscopy counting and with chemical analysis of AZAs. Whereas *Az. poporum* occurred only in very low densities at a few stations, higher abundances of *Az. spinosum* were detected along the Irish coastline and therefore underline the importance of current monitoring programs on that species performed by the Marine Institute (Galway, Ireland). *Amphidoma languida* revealed highest cell densities in the central North Sea, indicating this may be the dominant toxigenic amphidomatacean species in the North Sea area. However, this species was also detected at many stations in Irish waters and its surveillance should thus be incorporated into the Irish, as well as EU monitoring programs.

The on-board availability and (almost) real time operation of three approaches and instruments—microscopy, LC-MS/MS and qPCR—was successful and enabled the validation and comparison of results from different perspectives. In combination, the multi-method approach yielded sound and reliable data on diversity, distribution and abundance of toxigenic Amphidomataceae in the area.

## Supporting information

**S1 Table. Mass transitions m/z (Q1>Q3 mass) and their respective AZA.**
(PDF)

**S2 Table. Species counts (cells $L^{-1}$) based on qPCR (cell number calculation based on mean $C_T$ values of three technical replicates) and microscopy, as well as toxin amount (pg $L^{-1}$) based on LC-MS/MS for each station.** n.d. = not detected; n.a. = no data available.
(PDF)

**S1 Fig.** Geographical distribution of temperature (A), salinity (B) and fluorescence (C) averaged for the upper layer defined as the maximum sampling depth at each station.
(TIF)

**S2 Fig.** Depth profiles of temperature (A), salinity (B) and fluorescence (C) for the six defined sampling transects T1 –T6 as indicated in Fig 1B.
(TIF)

## Acknowledgments

The authors thank Caroline Cusack, Paula Hynes and Joe Silke (Marine Institute, Galway, Ireland), as well as Luisa Hintze and Karina Krapf (AWI, Bremerhaven, Germany) for on-board technical support on sampling and chemical analyzes of AZA samples. Thanks go also to Rohan Henkel (ICBM, Wilhelmshaven, Germany) for on-board CTD operation, data formatting and processing. We further thank Captain Diecks and the entire crew on *RV Heincke* for support and assistance during the field sampling campaign in this study.

## Author Contributions

**Conceptualization:** Stephan Wietkamp, Bernd Krock, Urban Tillmann.

**Data curation:** Bernd Krock.

**Formal analysis:** Stephan Wietkamp.

**Funding acquisition:** Bernd Krock, Urban Tillmann.

**Investigation:** Stephan Wietkamp, Bernd Krock, Dave Clarke, Daniela Voß, Jane Kilcoyne, Urban Tillmann.

**Methodology:** Stephan Wietkamp, Dave Clarke, Rafael Salas.

**Project administration:** Bernd Krock, Urban Tillmann.

**Resources:** Dave Clarke.

**Supervision:** Bernd Krock, Urban Tillmann.

**Validation:** Stephan Wietkamp, Bernd Krock, Dave Clarke, Urban Tillmann.

**Visualization:** Stephan Wietkamp, Daniela Voß, Urban Tillmann.

**Writing – original draft:** Stephan Wietkamp, Urban Tillmann.

**Writing – review & editing:** Stephan Wietkamp, Bernd Krock, Dave Clarke, Daniela Voß, Rafael Salas, Jane Kilcoyne, Urban Tillmann.

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
