## [Decision Letter · Decision Letter 0]

15 May 2020

PONE-D-20-07572

Distribution and abundance of azaspiracid-producing dinophyte species and their toxins in North Atlantic and North Sea waters in summer 2018

PLOS ONE

Dear Dr. Tillmann,

Thank you for submitting your manuscript to PLOS ONE. After careful consideration, we feel that it has merit but does not fully meet PLOS ONE’s publication criteria as it currently stands. Therefore, we invite you to submit a revised version of the manuscript that addresses the points raised during the review process.

ACADEMIC EDITOR: Be sure to:

two referees indicate only minor modifications, and I agree with them.

We would appreciate receiving your revised manuscript by Jun 29 2020 11:59PM. To enhance the reproducibility of your results, we recommend that if applicable you deposit your laboratory protocols in protocols.io, where a protocol can be assigned its own identifier (DOI) such that it can be cited independently in the future. For instructions see: http://journals.plos.org/plosone/s/submission-guidelines#loc-laboratory-protocols

We look forward to receiving your revised manuscript.

Kind regards,

Raffaella Casotti

Academic Editor

PLOS ONE

Additional Editor Comments (if provided):

dear Author, two reviewers found your manuscript interesting and worth publishing but only after minor modification and I agree with them.

2. In your Methods section, please provide additional location information of the sampling sites, including geographic coordinates for the data set if available.

3. In your Methods section, please provide additional information regarding the permits you obtained for the work. Please ensure you have included the full name of the authority that approved the sampling site access and, if no permits were required, a brief statement explaining why.

5. We note that Figures #1, 4, 6 and S3 in your submission contain map images which may be copyrighted. All PLOS content is published under the Creative Commons Attribution License (CC BY 4.0), which means that the manuscript, images, and Supporting Information files will be freely available online, and any third party is permitted to access, download, copy, distribute, and use these materials in any way, even commercially, with proper attribution. For these reasons, we cannot publish previously copyrighted maps or satellite images created using proprietary data, such as Google software (Google Maps, Street View, and Earth). For more information, see our copyright guidelines: http://journals.plos.org/plosone/s/licenses-and-copyright.

1.    You may seek permission from the original copyright holder of Figures #1, 4, 6 and S3 to publish the content specifically under the CC BY 4.0 license. 

Reviewers' comments:

Reviewer's Responses to Questions

**Comments to the Author**

1. Is the manuscript technically sound, and do the data support the conclusions?

Reviewer #1: Yes

Reviewer #2: Yes

2. Has the statistical analysis been performed appropriately and rigorously? 

Reviewer #1: Yes

Reviewer #2: Yes

3. Have the authors made all data underlying the findings in their manuscript fully available?

Reviewer #1: Yes

Reviewer #2: Yes

4. Is the manuscript presented in an intelligible fashion and written in standard English?

Reviewer #1: Yes

Reviewer #2: No

5. Review Comments to the Author

Reviewer #1: The manuscript reports results of an investigation aimed to i) detect and quantify species of the genera Azadinium and Amphidoma by the analysis of live phytoplankton samples and by qPCR and ii) detect azaspiracid toxins along 7 oceanographic transects around Ireland and in the North Sea sampled during a cruise carried out in July-August 2018. Three toxic species, Azadinium spinosum, Azadinium poporum and Amphidoma languida) were detected at several stations, where also various azaspiracid analogues could be quantified by LC-MS/MS. The qPCR assays are currently in use to detect the toxic Azadinium spinosum, Azadinium poporum in Ireland. However, the authors point out at the need for recurrent validations of the method that should be applied to a broad range of strains from different geographic areas in order to test its specificity and its use for the quantification of the species of interest (different populations may differ in the number copies of ribosomal DNA). To address this requirement, the authors report the results of validation tests of the qPCR assay using newly established strains of Az. spinosum and Az. poporum and also performed ‘spike experiments’ to test the reliability of the method to quantify cell abundances.

Azaspiracid toxins have been described relatively recently, and they constitute a threat for aquaculture in several countries, including Ireland. The species producing these toxins are very difficult to identify in light microscopy and therefore different methods should be used to detect their presence in natural samples. This manuscript represents a valuable contribution not only because it adds novel information on the geographic distribution of these toxic species but also because it shows the reliability of qPCR assays and the usefulness of combined and complementary approaches to detect the species and their toxins. The manuscript is clearly written, illustrations are appropriate and the results are thoroughly discussed.

I only have a few minor comments.

Line 71 and elsewhere (e.g. on lines 103, 104, 105, 135, 245…): When referring to the authors of a paper, list only the first author followed by ‘et al. ’which were more recently discovered by Satake et al. (8),

Line 89: suggested change: (which, together with Azadinium, ARE INCLUDED IN THE FAMILY Amphidomataceae),

Lines 180-181: ‘…values of the respective maximum sampling depth for temperature, salinity and fluorescence of the upper water layer were averaged.’ Can the authors specify what is the ‘upper water layer’? Is it the layer above the thermocline? See also the legend of Supplementary Fig. 3: ‘averaged for the upper layer defined as the maximum sampling depth at each station’: does this mean that the average data are all data from all sampled depths?

Lines 584-586: ‘While Amphidoma was much more abundant at station 71 compared to 72, fluorescence was much higher at station 72, which may indicate that different bloom development stages were sampled.’ The sentence is not very clear: I guess that Amphidoma was only a minor component of the whole phytoplankton assemblage and the higher fluorescence recorded at station 72 indicates that other phytoplankton species were abundant at this station

.

Legend of Supplementary Table 2: suggested change ‘….as well as toxin concentration….’

Reviewer #2: Review of: Distribution and abundance of azaspiracid-producing dinophyte species and their

toxins in North Atlantic and North Sea waters in summer 2018

In this study, a thorough investigation was conducted of Azadinium and closely related species during a cruise in Irish waters and the North Sea during 2018. The study used LCMS chemical detection, light microscopy, and qPCR to quantify two species of Azadinium and determine the concentration of Azaspiracids at multiple depths at sites along the transects examined.

The light microscopy, toxin analysis, and qPCR are thoroughly conducted, validated and the figures are very well presented and clear. I believe this study is a valuable contribution and should be published.

My comments mainly concern the English and the way in which the manuscript is written. I believe the manuscript needs to be thoroughly revised by the native speaker authors, to improve the language and flow of the text.

These are some examples from the Introduction:

Introduction

Lines 55-56: “…are one of the major threats for human health….” This is not correct as written, as harmful algae are not a major threat for human health as compared to every other current human health threat. Needs rewording.

Line 56: “(so-called blooms) – delete this

Line 57: delete the comma after both

Line 62: change ‘economy’ to ‘economic’

Line 65: change ‘phytoplanktonic’ to ‘phytoplankton’

Lines 66-69: delete the e.g. before each species name.

Line 71 : change Satake, Ofuji to Satake et al. This issue is the same throughout the whole manuscript, and needs to be checked and revised.

These are just some examples, but they illustrate the issues with language in the manuscript and the need for it to be checked.

Figure 5. This figure doesn’t contain any estimates of counting error for the two different methods. Since these errors were determined, ie replicate samples or counts, or known counting errors, then it would be useful to add these numbers to the figure.

6. PLOS authors have the option to publish the peer review history of their article (what does this mean?). If published, this will include your full peer review and any attached files.

Reviewer #1: Yes: Marina Montresor

Reviewer #2: No

---

## [Author Response · Author response to Decision Letter 0]

4 Jun 2020

Additional Editor Comments (if provided):

dear Author, two reviewers found your manuscript interesting and worth publishing but only after minor modification and I agree with them.

REPLY: PLOSOne style requirements were checked and incorporated where needed.

2. In your Methods section, please provide additional location information of the sampling sites, including geographic coordinates for the data set if available.

REPLY: station number and position are included in S2 Table. We now added this info in the M&M section

3. In your Methods section, please provide additional information regarding the permits you obtained for the work. Please ensure you have included the full name of the authority that approved the sampling site access and, if no permits were required, a brief statement explaining why.

REPLY: we now added the full name of the authorities that provided permissions in the text 

REPLY: Oxygen data are included in the Pangaea data set (Krock B, Wisotzki A. Physical oceanography during HEINCKE cruise HE516. Alfred Wegener Institute, Helmholtz Centre for Polar and Marine Research, Bremerhaven,. PANGAEA. 2018:https://doi.org/10.1594/PANGAEA.896405), so this reference is now replacing “data not shown”

Moreover, we noted that we – in the discussion – used unpublished information of co-authors cited as (Kilcoyne et al, unpublished, 2 x Clarke et al, unpublished) three times. This is now changed as follows

- the statement based on unpublished data of Kilcyone et al is removed from the manuscript

- data referred to by citing “Clarke et al, unpublished” are in fact now “in press” in a conference proceeding, so on both places in the discussion this reference (which is now added to the reference list) was included.

5. We note that Figures #1, 4, 6 and S3 in your submission contain map images which may be copyrighted. All PLOS content is published under the Creative Commons Attribution License (CC BY 4.0), which means that the manuscript, images, and Supporting Information files will be freely available online, and any third party is permitted to access, download, copy, distribute, and use these materials in any way, even commercially, with proper attribution. For these reasons, we cannot publish previously copyrighted maps or satellite images created using proprietary data, such as Google software (Google Maps, Street View, and Earth). For more information, see our copyright guidelines: http://journals.plos.org/plosone/s/licenses-and-copyright.

REPLY: We now obtained via e-mail the written permission of the copyright holder of Ocean Data View (Prof. Dr. Reiner Schlitzer). The e-mail is now included as pdf file in the submission.

 

Reviewer #1: The manuscript reports results of an investigation aimed to i) detect and quantify species of the genera Azadinium and Amphidoma by the analysis of live phytoplankton samples and by qPCR and ii) detect azaspiracid toxins along 7 oceanographic transects around Ireland and in the North Sea sampled during a cruise carried out in July-August 2018. Three toxic species, Azadinium spinosum, Azadinium poporum and Amphidoma languida) were detected at several stations, where also various azaspiracid analogues could be quantified by LC-MS/MS. The qPCR assays are currently in use to detect the toxic Azadinium spinosum, Azadinium poporum in Ireland. However, the authors point out at the need for recurrent validations of the method that should be applied to a broad range of strains from different geographic areas in order to test its specificity and its use for the quantification of the species of interest (different populations may differ in the number copies of ribosomal DNA). To address this requirement, the authors report the results of validation tests of the qPCR assay using newly established strains of Az. spinosum and Az. poporum and also performed ‘spike experiments’ to test the reliability of the method to quantify cell abundances.

Azaspiracid toxins have been described relatively recently, and they constitute a threat for aquaculture in several countries, including Ireland. The species producing these toxins are very difficult to identify in light microscopy and therefore different methods should be used to detect their presence in natural samples. This manuscript represents a valuable contribution not only because it adds novel information on the geographic distribution of these toxic species but also because it shows the reliability of qPCR assays and the usefulness of combined and complementary approaches to detect the species and their toxins. The manuscript is clearly written, illustrations are appropriate and the results are thoroughly discussed.

I only have a few minor comments.

Line 71 and elsewhere (e.g. on lines 103, 104, 105, 135, 245…): When referring to the authors of a paper, list only the first author followed by ‘et al. ’which were more recently discovered by Satake et al. (8),

REPLY: To precisely follow the journal requirement for literature formatting we downloaded and used the endnote style file provided by on the journals home page. As this file obviously produce a wrong formatting we now changed naming authors in the text it in the whole manuscript.

Line 89: suggested change: (which, together with Azadinium, ARE INCLUDED IN THE FAMILY Amphidomataceae),

REPLY: Changed as suggested 

Lines 180-181: ‘…values of the respective maximum sampling depth for temperature, salinity and fluorescence of the upper water layer were averaged.’ Can the authors specify what is the ‘upper water layer’? Is it the layer above the thermocline? See also the legend of Supplementary Fig. 3: ‘averaged for the upper layer defined as the maximum sampling depth at each station’: does this mean that the average data are all data from all sampled depths?

REPLY: We agree that this was ambigously specified and we changed the sentence to:

“For field data correlations as well as visualization (S3 Figure), CTD profile data for temperature, salinity and fluorescence were averaged from the surface down to the maximum sampling depth.”

Lines 584-586: ‘While Amphidoma was much more abundant at station 71 compared to 72, fluorescence was much higher at station 72, which may indicate that different bloom development stages were sampled.’ The sentence is not very clear: I guess that Amphidoma was only a minor component of the whole phytoplankton assemblage and the higher fluorescence recorded at station 72 indicates that other phytoplankton species were abundant at this station.

Reply: We agree that the sentence was not entirely clear. To emphasize that variable densities of other species are important we changed the sentence which now is:

“While Amphidoma was much more abundant at station 71 compared to 72, fluorescence was much higher at station 72, which may indicate that different plankton communities or different development stages were sampled.”

Legend of Supplementary Table 2: suggested change ‘….as well as toxin concentration….’

REPLY: In a strict chemical definition “concentration” is only applicable to dissolved compounds and accordingly would refer to dissolved toxins in the seawater, which is not the parameter we report. For description of quantities of particles or particulate toxin content, we consistently prefer to use the terms “density”, “abundance”, or “amount”. 

 

Reviewer #2: Review of: Distribution and abundance of azaspiracid-producing dinophyte species and their toxins in North Atlantic and North Sea waters in summer 2018

In this study, a thorough investigation was conducted of Azadinium and closely related species during a cruise in Irish waters and the North Sea during 2018. The study used LCMS chemical detection, light microscopy, and qPCR to quantify two species of Azadinium and determine the concentration of Azaspiracids at multiple depths at sites along the transects examined. The light microscopy, toxin analysis, and qPCR are thoroughly conducted, validated and the figures are very well presented and clear. I believe this study is a valuable contribution and should be published.

My comments mainly concern the English and the way in which the manuscript is written. I believe the manuscript needs to be thoroughly revised by the native speaker authors, to improve the language and flow of the text.

REPLY: We, and especially the two native speaker co-authors from Ireland, again carefully revised the manuscript to improve the language and flow of text.

These are some examples from the Introduction:

Introduction

Lines 55-56: “…are one of the major threats for human health….” This is not correct as written, as harmful algae are not a major threat for human health as compared to every other current human health threat. Needs rewording.

REPLY: We agree the sentence now is:

“Marine toxic microalgae are of concern for human health and aquaculture industries worldwide.”

Line 56: “(so-called blooms) – delete this

REPLY: Agree, deleted

Line 57: delete the comma after both

REPLY: deleted

Line 62: change ‘economy’ to ‘economic’

REPLY: done

Line 65: change ‘phytoplanktonic’ to ‘phytoplankton’

REPLY: done

Lines 66-69: delete the e.g. before each species name.

REPLY: No. These are only examples of species producing these toxins, so e.g. is needed here.

Line 71 : change Satake, Ofuji to Satake et al. This issue is the same throughout the whole manuscript, and needs to be checked and revised.

REPLY: To precisely follow the journal requirement for literature formatting we downloaded and used the endnote style file provided by on the journals home page. As this file obviously produce a wrong formatting we now changed naming authors in the text it in the whole manuscript.

These are just some examples, but they illustrate the issues with language in the manuscript and the need for it to be checked.

Figure 5. This figure doesn’t contain any estimates of counting error for the two different methods. Since these errors were determined, ie replicate samples or counts, or known counting errors, then it would be useful to add these numbers to the figure.

REPLY: This comment is difficult to understand? Fig 5 show data obtained with the same “method” but for two different qPCR assays? In any case, replicate estimates for all qPCR assays are technical replicates from one sample per station, and cell number calculations for a station are based on the average Ct value of the technical replicate and thus no error bars in Fig. 5 are plotted. This specification now is explicitly added to the M&M section, in the legend of Fig. 5, and in the header of S2 Table.

---

## [Editor Report · Decision Letter 1]

8 Jun 2020

Distribution and abundance of azaspiracid-producing dinophyte species and their toxins in North Atlantic and North Sea waters in summer 2018

PONE-D-20-07572R1

Dear Dr. Tillmann,

We’re pleased to inform you that your manuscript has been judged scientifically suitable for publication and will be formally accepted for publication once it meets all outstanding technical requirements.

Kind regards,

Raffaella Casotti

Academic Editor

PLOS ONE

Additional Editor Comments (optional):

Dear Authors, I considered the changes applied satisfactory and therefore recommend the manuscript for publication. Congratulations
---

## [Editor Report · Acceptance letter]

10 Jun 2020

PONE-D-20-07572R1 

Distribution and abundance of azaspiracid-producing dinophyte species and their toxins in North Atlantic and North Sea waters in summer 2018 

Dear Dr. Tillmann:

I'm pleased to inform you that your manuscript has been deemed suitable for publication in PLOS ONE. Congratulations! Your manuscript is now with our production department. 

Kind regards, 

on behalf of

Dr. Raffaella Casotti 

Academic Editor

PLOS ONE